# Evaluation of Different Black Mulberry Fruits (*Morus nigra* L.) Based on Phenolic Compounds and Antioxidant Activity

**DOI:** 10.3390/foods11091252

**Published:** 2022-04-26

**Authors:** Ri-Si Wang, Pan-Hao Dong, Xi-Xiang Shuai, Ming-Shun Chen

**Affiliations:** State Key Laboratory of Food Science and Technology, Nanchang University, Nanchang 330047, China; 357213317016@ncu.edu.cn (R.-S.W.); 412314918140@ncu.edu.cn (P.-H.D.); 405604811146@ncu.edu.cn (X.-X.S.)

**Keywords:** mulberry fruit, cultivar, phenolic compounds, antioxidant activity

## Abstract

This study evaluated thirteen different black mulberry fruits (*Morus nigra* L.) grown in the Guangdong region in order to select the best cultivar for health benefits and commercial applications. The phenolic compounds were identified and quantified using UPLC-ESI-MS/MS. The antioxidant activity was evaluated by three in vitro methods. Significant differences among samples were found regarding total soluble solids (6.20–15.83 °Brix), titratable acidity (5.82–48.49 mg CA/g), total phenolic contents (10.82–27.29 mg GAE/g), total flavonoid contents (1.21–2.86 mg RE/g) and total anthocyanin contents (2.91–11.86 mg CE/g). Fifty-five different phenolic compounds were identified, of which fifteen were reported in mulberry for the first time, but only forty-six of them were quantitated. The DPPH radical scavenging activity, ABTS radical scavenging activity and ferric ion-reducing antioxidant power varied significantly among the samples. Overall, cultivars with better combinations of phenolic compounds and antioxidant activity were Qiong46 (M-2), Yuebanguo (M-4) and Heizhenzhu (M-10), which were recommended for commercial cultivation.

## 1. Introduction

Mulberry, belonging to the *Morus* genus of the *Moraeeae* family, is an edible, medicinal and economical woody plant that has been cultivated by humans for thousands of years. Mulberries grow in an extensive range of climatic, geographical and soil conditions, and are widely distributed around the world [1]. In the past, mulberry fruit was regarded as a by-product of the sericulture industry in China. With the growing demand for high-quality berry fruits, mulberry fruit has become a popular fruit nowadays due to its high nutritional value. Several investigations revealed that mulberry fruits possess a wide scope of biochemical activities, such as antioxidant, anti-hyperlipidemia and anti-cancer properties, due to their rich nature phenolic compounds, including phenolic acids, flavonols and anthocyanins [2,3,4]. However, the chemical composition and content of phenolic compounds of different mulberry cultivars vary greatly [5,6]. There are three mulberry species produced in China, namely white mulberry (*Morus alba* L.), red mulberry (*Morus rubra* L.) and black mulberry (*Morus nigra* L.) [7]. Compared with the white and red mulberry fruit, the black mulberry fruit is gaining increasing attention due to its attractive color and higher content of phenolic compounds, especially anthocyanin compounds [1,8].

In order to satisfy the market demand for mulberry fruit, decades of breeding programs have cultivated many new cultivars to improve the commercial quality of the fruit, such as yield, size, taste and disease resistance [9]. To the best of our knowledge, many improved black mulberry cultivars, such as Jinqiang63, Yunguo1, Yichuanhong, Yuebanguo, Qiong46, Yueshen28, Heizhenzhu, Xuanguo1, Guiyou12 and Shansang, have not been reported. Gaining knowledge of the phenolic composition of new black mulberry fruits remains an interesting and important task. To this end, thirteen different black mulberry cultivars from the same region were systematically investigated. The total soluble solids, titratable acidity, total phenolic content, total flavonoid content and total anthocyanin content in black mulberry fruits were determined. Identification and quantification of polyphenols were realized using UPLC-ESI-MS/MS. The antioxidant activities of the mulberry extracts were evaluated by DPPH, ABTS and FRAP assays. The purpose of this study was to compare different cultivars of mulberry fruits to select cultivars with good properties for commercial cultivation.

## 2. Materials and Methods

### 2.1. Plant Materials

Thirteen cultivars of black mulberry fruits (*M. nigra* L.) including Yichuanhong (M-1), Qiong46 (M-2), Guoshang8632 (M-3), Yuebanguo (M-4), Yueshen28 (M-5), Yueshen74 (M-6), Da10 (M-7), Jinqiang63 (M-8), Yunguo1 (M-9), Heizhenzhu (M-10), Xuanguo1 (M-11), Guiyou12 (M-12) and Shansang (M-13) were collected from the Chinese Academy of Tropical Agricultural Science (Zhanjiang, China) in April 2019. Mulberry fruits of similar size and color were randomly picked from different parts of the plant at the commercially full maturity stage and transported to the lab. After washing, the fruits were stored in the refrigerator at −20 °C until analysis.

### 2.2. Chemicals

Sodium hydroxide, monobasic potassium phosphate, ascorbic acid and Folin-Ciocalteu reagent were purchased from Aladdin (Shanghai, China), and 2,2 diphenyl-1-picrylhydrazyl radical (DPPH), 2-2′azino-bis-(3 ethylbenzothiazoline-6-sulphonic acid) diammonium salt (ABTS), 2,4,6 tripyridy-S-triazine (TPTZ) and phenolic compound standards were obtained from Sigma-Aldrich Chemical Co (Shanghai, China). Methanol and acetonitrile of LC grade were provided by Aladdin (Shanghai, China). Ultrapure water was prepared with a Milli-Q system (Millipore, Billerica, MA, USA).

### 2.3. The Total Soluble Solids and Titratable Acidity

The total soluble solids (TSS) and titratable acidity (TA) of mulberry fruits were determined by official standard procedures [10]. The TSS was measured by a digital refractometer (PAL-1, Atago, Koriyama, Japan) after dripping one drop of fruit juice into it. The TA was measured by the titration method with sodium hydroxide standard solution and expressed as mg citric acid (CA)/g fresh weight (FW).

### 2.4. Phenolic Compound Extraction

The phenolic compounds were extracted according to a previous report [11]. The mulberry fruit was first pre-frozen at below −40 °C for 24 h and then freeze-dried at −50 °C, 7.23 Pa for 24 h by a vacuum freeze dryer (FreeZone 4.5L, Labconco, KS, USA) and milled into powder in a small grinder (HR-10, Shanghai haris electric appliance Co., Ltd., Shanghai, China). A total of 800 mg of mulberry fruit powder was extracted using 4 mL of methanol/water (80:20, *v*/*v*) acidified with hydrochloric acid (0.5%). Then, the mixture was sonicated at 125 W for 25 min and centrifuged at 8000 g/min for 5 min at room temperature. The extraction procedure was repeated thrice. The supernatants were concentrated at 45 °C by a rotary evaporator (RE 2000A, Yarong biochemistry instrument factory, Shanghai, China) to obtain the extracted fraction. Then, the extracted fraction was dissolved in distilled water and chromatographed through a macroporous resin D101 column (25 cm × 16 mm), which was eluted with 60% ethanol. The eluent was concentrated at 45 °C in vacuum and freeze-dried for further analysis.

### 2.5. Total Phenolic Content, Flavonoid Content and Anthocyanin Content

The total phenolic content (TPC) was determined by the Folin-Ciocalteu method [12]. Briefly, 1.0 mL of the sample was mixed with 0.5 mL of Folin-Ciocalteu reagent (0.2 M), and then 2.0 mL of 7.5% Na_2_CO_3_ and 6.5 mL distilled water were added. The mixture was incubated at 70 °C for 30 min in the dark, and the absorbance was measured at 750 nm by a microplate reader (Spark 10M, Tecan, Switzerland). TPC was expressed as mg gallic acid equivalents (GAE) per g of dry weight (DW).

The total flavonoid content (TFC) was measured by the aluminum chloride colorimetric method [13]. From each sample, 0.5 mL was mixed with 0.3 mL of 5% NaNO_2_ for 6 min at room temperature, and then 0.3 mL of 10% AlCl_3_ was added. After 6 min, 4 mL of 4% NaOH and 2.4 mL distilled water were added. The mixture was kept for 15 min at room temperature, and the absorbance was recorded against the blank at 510 nm by a microplate reader. The results were expressed as mg rutin equivalents (RE) per g of DW.

The total anthocyanin content (TAC) was determined using the pH differential method [14]. The sample solution was diluted with 0.025 M potassium chloride buffer (pH_1.0_) and 0.2 M sodium acetate buffer (pH_4.5_). The absorbance was measured at 510 and 700 nm by a microplate reader. The TAC expressed as mg cyanidin-3-*O*-glucoside equivalents (CE) per g of DW was calculated as follows:
TAC = (A/ε × L) × MW × DF × V/DW × 1000(1)
A = (A_510_ − A_700_) pH_1.0_ − (A_510_ − A_700_) pH_4.5_(2)
where A is absorbance, MW is cyanidin-3-*O*-glucoside molecular weight (449.2), ε is the cyanidin-3-*O*-glucoside molar absorptivity (26,900), DF is the dilution factor, V is the last volume, DW is the dry weight of mulberry and L is the optical path (1 cm).

### 2.6. Identification and Quantification of Polyphenols

UPLC analyses were performed using an AB SCIEX UPLC system connected to a Qtrap 6500+ Mass spectrometer with electrospray ionization (ESI, AB SCIEX, Framingham, MA, USA). The separation was performed at 25 °C on a Waters UPLC HSS T3 column (100 mm × 2.1 mm, 1.7 μm, Waters, MD, USA). The mobile phase was water with 0.1% formic acid (*v*/*v*) (A) and acetonitrile (B). The linear gradient program was as follows: 0–2 min 100% A, 2–36 min 100–45% A, 36–39 min 45–5% A, 39–42 min 5% A, 42–42.1 min 5–100% A and 42.1–45 min 100% A. The flow rate was 0.3 mL/min, and the injection volume was 5 μL.

The MS analysis was performed in both negative ionization mode and positive ionization mode. The MS spectra were recorded from *m*/*z* 50 to 1000. The ESI parameters in negative ionization mode were as follows: atomizing gas (GS1) 60 psi, auxiliary gas (GS2) 50 psi, curtain gas (CUR) 35 psi, entrance potential (EP) 10 eV, collision cell entrance potential (CXP) 10 eV, ion source temperature (TEM) 600 °C, ion spray voltage floating (ISVF) −4500 V and collision energy (CE) −35 eV. The ESI parameters in positive ionization mode were as follows: GS1 60 psi, GS2 50 psi, CUR 35 psi, EP 10 eV, CXP 10 eV, TEM 600 °C, ISVF 5500 V and CE −35 eV. All data were acquired and processed using SCIEX OS-MQ software (AB Sciex, Framingham, MA, USA).

Mulberry phenolic compounds were identified according to their molecular weight (mass spectra), MS/MS fragmentations and information as reported in the previous literature. A series of standard solutions with concentrations of 0.05 ppb, 0.2 ppb, 1 ppb, 3 ppb, 10 ppb, 25 ppb, 50 ppb, 100 ppb and 200 ppb were prepared. The multiple reaction monitor (MRM) was applied for quantitative detection based on accurate molecular weight. Phenolic compounds were quantified using the calibration curve of phenolic compound standards.

### 2.7. Antioxidant Activity

#### 2.7.1. DPPH Radical Scavenging Activity

The DPPH radical scavenging activity was determined according to the literature [15], with some modifications. Briefly, 10 μL of sample solution was mixed with 200 μL of 0.066 mM DPPH solution and reacted in the dark for 30 min. The absorbance was measured at 515 nm using a microplate reader. Vitamin C was used as a positive control. The DPPH radical scavenging activity was expressed as mg vitamin C equivalent (VCE) per g of DW.

#### 2.7.2. ABTS Radical Scavenging Activity

The ABTS radical scavenging activity was carried out according to the method described in the literature [16], with slight modifications. Briefly, ABTS stock solution was prepared by adding 10 mL of 7.0 mM of 2,2-azobis (2-amidinopropane) dihydrochloride to 5.0 mL of 2.45 mM aqueous potassium persulfate. The ABTS stock solution was incubated for 12 h at room temperature and diluted with ethanol to obtain an absorbance of 0.70 ± 0.05 at 750 nm. A total of 10 μL of sample solution was mixed with 200 μL of diluted ABTS solution. The mixture was reacted for 6 min and read at 750 nm. The ABTS radical scavenging activity was expressed as mg VCE per g of DW.

#### 2.7.3. Ferric Ion Reducing Antioxidant Power (FRAP)

FRAP was performed as described in the literature [17]. The FRAP reagent was prepared by mixing 10 mM TPTZ solution (in 40 mM HCL), 20 mM FeCL_3_ and 300 mM acetate buffer (pH 3.6) in proportion 1:1:10 (*v*/*v*/*v*), respectively. Then, the mixture was warmed at 37 °C for 30 min. A total of 10 μL of sample solution was mixed with 200 μL FRAP solution. Then the mixture was incubated at 37 °C for 30 min, and the absorbance was read at 595 nm. The FRAP was expressed as mg VCE per g of DW.

### 2.8. Statistical Analysis

All samples were prepared in triplicate. Data of all results are expressed as mean ± standard deviation (SD). Statistical analyses were performed with software SPSS 16.0 (SPSS Inc., Chicago, IL, USA). Correlation analysis was carried out by Pearson’s test. Principal component analysis (PCA) and hierarchical cluster analysis (HCA) were calculated using SIMCA 14.1 (Umetrics AB, Umea, Sweden).

## 3. Results and discussion

### 3.1. The Total Soluble Solids and Titratable Acidity

TSS and TA are basic qualities that affect the sensory quality and marketability of fresh fruits, and they are related to the level of soluble sugars and organic acid in the fruits, respectively [18]. As shown in Table 1, the contents of TSS and TA varied considerably among cultivars. TSS content of mulberry fruits ranged from 6.20 (M-10) to 15.83 °Brix (M-12), which was in agreement with that of mulberry fruits from Zhejiang region of China (6.2–16.00 °Brix) [9], but lower than that of mulberry fruits grown in Spain (12.00–25.80 °Brix) [18]. The level of TA ranged between 5.82 mg CA/g FW and 48.49 mg CA/g FW. Sample M-8 contained the highest TA value (48.49 mg/g FW), which was similar to the value of the black mulberry cultivar Yaosang, grown in the Xinjiang region of China (47.10 mg/g FW) [8]. Sample M-3 showed the lowest TA value (5.82 mg CA/g FW).

### 3.2. The Total Phenolic Content, Total Flavonoid Content and Total Anthocyanin Content

The TPC, TFC and TAC of mulberry fruits were shown in Table 1. TPC ranged from 10.82 mg GAE/g DW (M-3) to 27.29 mg GAE/g DW (M-10); TFC varied from 1.21 RE/g DW (M-3) to 2.86 mg RE/g DW (M-6); and TAC ranged between 2.91 CE/g DW (M-3) and 11.86 mg CE/g DW (M-2). Butkhup et al. [19] showed that TFC of Thailand mulberry fruits ranged from 0.70 mg RE/g DW to 2.11 mg RE/g DW, which was similar to our results. Samples M-2, M-4 and M-10 contained high TPC, TFC and TAC values, which were significantly higher than those of mulberry fruits cultivated in the Zhenjiang region of China (1.59–2.46 mg GAE/g DW, 0.41–1.50 mg RE/g DW and 0.19–1.93 mg CE/g DW, respectively) [6]. Bae and Suh [4] reported that TPC, TFC and TAC of different mulberry clones in Korea were 2.24–2.57 mg GAE/g DW, 0.02–0.07 mg/g DW and 1.23–2.06 mg/g DW, respectively, which were lower than those of this study. Furthermore, TPC, TFC and TAC of samples M-2, M-4, M-6 and M-10 were several times higher than those of sample M-3. These cultivars with higher values of TPC, TFC and TAC might be more suitable as abundant sources of natural phenolic compounds. Considering that the investigated mulberry cultivars were collected from the same growth environment, the difference in the contents of TPC, TFC and TAC may be affected by genetic variation.

### 3.3. Identification and Quantification of Individual Polyphenols

Fifty-five different phenolic compounds were identified in mulberry cultivars based on their retention times and the fragmentation patterns of MS and MS^2^ mass spectra, of which fifteen compounds (compounds 12, 15, 19, 21, 22, 24, 27, 37, 38, 39, 41, 43, 44, 47 and 53) were reported in mulberry for the first time (Table 2). Meanwhile, forty-six phenolic compounds were quantified by the calibration curve of phenolic compound standards (Table 3).

Phenolic acid was the dominant phenolic compound in mulberry fruits, and fourteen phenolic acids were identified and quantified. Among them, sinapic acid was detected in mulberry fruits for the first time. Chlorogenic acid and cryptochlorogenic acid were the main phenolic acids, accounting for more than 85% of the total phenolic acid content in mulberry fruits, which was in line with the findings of Chu et al. [20] and Memon et al. [21], but contradict with some previous studies ([7,22,23], which reported that flavonoid was the main phenolic compound in mulberry fruits. These differences may be due to discrepancies in fruit maturity, genetics and environmental conditions.

Regarding flavonoids, twenty-two compounds belonging to five subclasses (flavonols, flavanones, flavanols, flavones and dihydrochalcones) were identified and quantified. As far as we know, dihydromyricetin, narcissin, aromadendrin, prunin, hesperidin, butein and trilobatin have never been described in mulberry fruits previously. The major flavonoids in mulberry fruits were flavonols mainly from rutin, kaempferol-3-O-rutinoside and quercetin, as well as flavones mainly from morin, which was consistent with the report of Sanchez-Salcedo et al. [11].

Anthocyanins play a prominent role in the mulberry fruit because they are directly related to color and possess varieties of biological activities [24]. Five anthocyanins were identified and quantified by UPLC-ESI-MS/MS. Cyanidin-3-*O*-rutinoside, delphinidin-3-O-glucoside and cyanidin-3,5-diglucoside were the dominant anthocyanins, accounting for more than 99% of total anthocyanin content, which was consistent with the results reported by Bao et al. [25], who found that cyanidin-3-O-rutinoside was the main anthocyanin. Pelargonidin and pelargonidin-3-O-glucoside accounted for a low proportion, and pelargonidin-3-*O*-glucoside was only found in cultivars M-3, M-5 and M-7.

In summary, the profile of phenolic compounds varied considerably in different cultivars. The contents of total individual phenolic compounds ranged between 292.16 mg/kg DW and 1672.21 mg/kg DW. There was no correlation between the content of total phenolic compounds quantified by HPLC and Folin-Ciocalteu method. These differences could be explained by the fact that not all phenolic compounds present in the samples were quantified by HPLC. The Folin-Ciocalteu method is an estimate and may be overestimated [26]. Samples M-2, M-4 and M-10 contained the highest content of individual phenolic compounds, and the lowest content of individual phenolic compounds was found in M-3, which was consistent with TPC.

### 3.4. Antioxidant Capacity

Antioxidants could combine with free radicals through hydrogen atoms to form very stable products, so they can prevent the chain reaction of free radicals to exert antioxidant effects. Owing to the complexity of the extracted components and the involvement of different reaction mechanisms, a combination of assays (DPPH, ABTS and FRAP) was used to evaluate the antioxidant capacity of mulberry fruits [27].

As shown in Figure 1, the antioxidant capacity varied significantly among different samples (*p* < 0.05). The antioxidant capacities of mulberry fruits determined by DPPH, ABTS and FRAP assays were 28.48–100.11 mg VCE/g DW, 13.49–73.06 mg VCE/g DW and 10.05–48.35 mg VCE/g DW, respectively. Samples M-2 and M-10 possessed higher antioxidant activity than other cultivars, while the lowest antioxidant activity was found in samples M-3, M-5 and M-12. Bao et al. [25] report the antioxidant capacities of white mulberry fruits as determined by DPPH, ABTS and FRAP assays (0.51–0.73 mg VCE/g DW, 0.07–0.08 mg VCE/g DW and 0.12–0.16 mg VCE/g DW, respectively) were significantly lower than those of samples M-2 and M-10 in this study, which might be due to the fact that white mulberry fruits contain lower levels of polyphenols (0.85–0.94 mg GAE/g DW) than black mulberry fruits.

As shown in Table 4, a significant correlation between the TPC, TFC, TAC and antioxidant activity was observed (*p* < 0.05), which was greatly in line with the previous findings that TPC, TFC, TAC and antioxidant activity showed a significant correlation [28,29]. However, some samples with similar TPC, TFC and TAC did not exhibit similar antioxidant activities. A possible explanation is that in addition to polyphenols, other non-phenolic compounds (including amino acids, enzymes and organic acids) also contribute to antioxidant activity. Moreover, the probable interactions among the different antioxidant compounds also play an important role in the overall antioxidant capacity [9].

### 3.5. Principal Components Analysis and Hierarchical Cluster Analysis

PCA is a statistical method widely used to discover relationships between variables. In order to understand the trends and relationships between many variables among different mulberry cultivars, PCA was applied to TSS, TA, TPC, TFC, TAC, DPPH, ABTS and FRAP and quantified individual phenolic compounds. The PCA resulted in a two-component model that explained 68.09% of the total variance. The first principal component accounted for 54.19%, and the second component explained 13.90% of the total variance. The PC score plots of the samples showed the classification of 13 mulberry cultivars (Figure 2A). The loading plots of the sample were shown in Figure 2B. PC1 was positively correlated with the contents of the individual phenolic compounds, TA, TPC, TFC, TAC, DPPH, ABTS and FRAP. PC2 was positively correlated with the variables of TSS, trans-cinnamic acid, rutin, sinapic acid, fraxin, delphinidin-3-*O*-glucoside and prunin.

Cultivars M-2, M-4 and M-10 showed high positive PC1 values, corresponding to high phytochemical contents (TPC, TFC, TAC and polyphenols) and antioxidant activity, as well as low TSS. Cultivars M-1, M-6, M-7 and M-9 exhibited similar PC1 and PC2 values, characterized by high TA. Cultivars M-3, M-5, M-8, M-11, M-12 and M-1 presented negative PC1 values, characterized by high TSS as well as low phytochemical contents and antioxidant activity. In addition, among all the quantitative phenolic compounds, according to their closeness in the loadings plot, eriodictyol and luteolin could be considered to have the greatest impact on antioxidant activity (DPPH, ABTS and FRAP assays).

HCA is a classification method based on the analyzed variables which investigates a natural grouping between samples by the values of measured features. All data of variables are standardized and processed using the square Euclidean distance and centroid clustering method, which calculates the similarity between different categories of data points to construct a dendrogram (Figure 3). The results exhibit a clear clustering tendency of mulberry samples, which contain similar profiles to the investigated data. All mulberry cultivars are divided into three main parts when the 50-distance threshold is chosen, based on the contents of TSS, TA, TPC, TFC, TAC, ABTS, DPPH, FRAP and individual phenolic compounds. The first group, consisting of M-2, M-4 and M-10, exhibits low levels of TSS, medium TA and high TPC, TFC, TAC, individual polyphenol contents and antioxidant activities. The second group, comprising M-1, M-6, M-7 and M-9, has high TA and medium TPC, TFC, TAC, individual polyphenol contents and antioxidant activities. The third group, including M-3, M-5, M-8, M-11, M-12 and M-13, shows high values of TSS, medium TA and low TPC, TFC, TAC, individual polyphenol contents and antioxidant activities. In summary, we conclude that different black mulberry cultivars can be completely distinguished by chemical characteristics. These results provide valuable information for future biological studies of mulberry fruits.

## 4. Conclusions

In this study, the phenolic compounds and antioxidant capacity of 13 different black mulberry fruits were compared and characterized. The TSS, TA, TPC, TFC, TAC and antioxidant activities of different cultivars of mulberry fruits were rather variable. The TPC, TFC and TAC of mulberry fruits were positively correlated with their antioxidant activities. The UHPLC-ESI- MS/MS analysis identified 55 phenolic compounds. Among them, 15 were reported for the first time in mulberry. Cultivars M-2, M-4 and M-10 are distinguished by their high TPC, TFC and TAC, strong antioxidant activities, as well as high levels of eriodictyol and luteolin, and they are recommended for clonal propagation and commercialization. Our results are helpful for selecting mulberry cultivars with abundant phenolic compounds as dietary supplements, food natural antioxidants as well as functional foods. However, more detailed biological studies are still needed to further elucidate the health benefits of these new mulberry cultivars.

## Figures and Tables

**Figure 1 foods-11-01252-f001:**
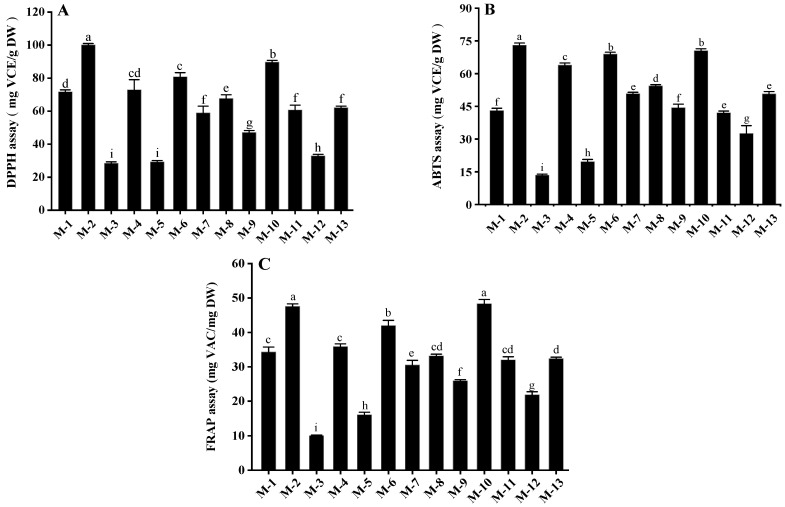
Antioxidant activities of different mulberry samples measured by (**A**) DPPH, (**B**) ABTS and (**C**) FRAP assays. Values in the different letters are significant differences (*p* < 0.05). M-1, Yichuanhong; M-2, Qiong46; M-3, Guosang8632; M-4, Yuebanguo; M-5, Yueshen28; M-6, Yueshen74; M-7, D10; M-8, Jinqiang63; M-9, Yunguo1; M-10, Heizhenzhu; M-11, Xuanguo1; M-12, Guiyou12; M-13, Shansang.

**Figure 2 foods-11-01252-f002:**
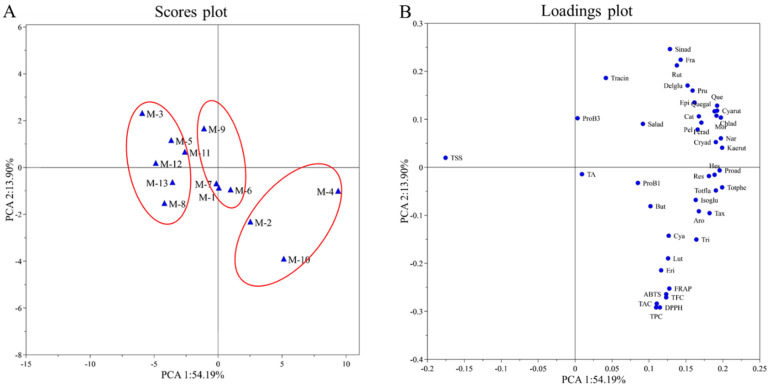
(**A**) PC scores plot of the mulberry samples; (**B**) loadings plot of the mulberry samples. Salad, salicylic acid; TraCin, trans-cinnamic acid; Proad, protocatechuic acid; Chlad, chlorogenic acid; Cryad, cryptochlorogenic acid; Ferad, ferulic acid; Sinad, sinapic acid; Nar, narcissin; Aro, aromadendrin; Tax, taxifolin; Quegal, quercetin-3-galactoside; Isoglu, isorhamnetin-3-O-glucoside; Kaerut, kaempferol-3-O-rutinoside; Rut, rutin; But, butein; Eri, eriodictyol; Pru, prunin; Hes, hesperidin; Tri, trilobatin; Lut, luteolin; Mor, morin; Que, quercetin; Cat, catechin; Epi, epicatechin; Pel, pelargonidin; Delglu, delphinidin-3-glucoside; Cya, cyanidin-3,5-diglucoside; Cyarut, cyanidin-3-O-rutinoside; ProB1, procyanidin B1; ProB3, procyanidin B3; Fra, fraxin; Res, resveratrol.

**Figure 3 foods-11-01252-f003:**
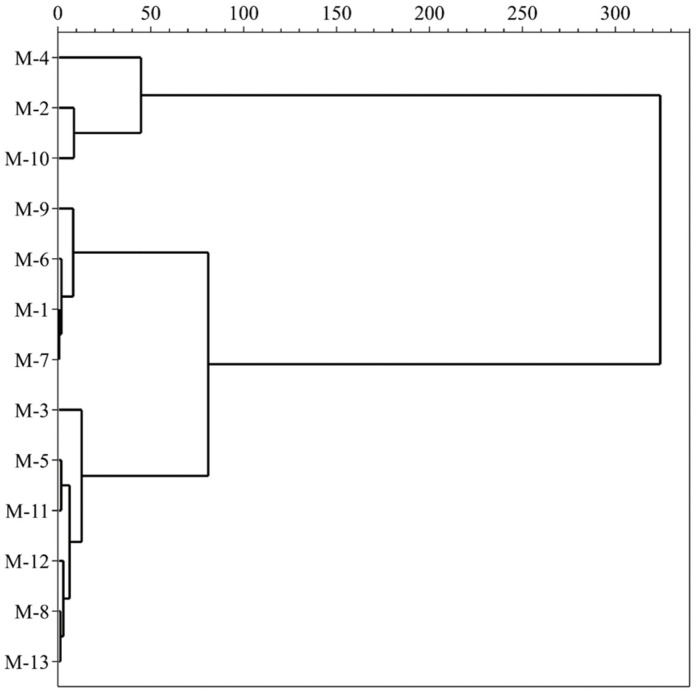
Hierarchical cluster analysis of the mulberry samples.

**Table 1 foods-11-01252-t001:** The total soluble solids (TSS), the titratable acidity (TA), total phenolic contents (TPC), total flavonoid contents (TFC) and total anthocyanin contents (TAC) of mulberry samples.

Cultivar	TSS (°Brix)	TA (g CA/kg FW)	TPC (mg GAE/g DW)	TFC (mg RE/g DW)	TAC (mg CE/g DW)
M-1 (Yichuanhong)	13.60 ± 0.49 b	10.05 ± 0.29 g	18.74 ± 0.31 g	1.99 ± 0.03 e	8.33 ± 0.05 e
M-2 (Qiong46)	12.30 ± 0.57 cd	26.82 ± 0.58 c	26.10 ± 0.06 b	2.80 ± 0.02 ab	11.86 ± 0.08 a
M-3 (Guosang8632)	13.62 ± 0.05 b	5.82 ± 0.41 i	10.82 ± 0.26 j	1.21 ± 0.07 h	2.91 ± 0.05 j
M-4 (Yuebanguo)	8.13 ± 0.61 e	7.19 ± 0.14 i	25.18 ± 0.17 c	2.75 ± 0.01 b	8.53 ± 0.14 d
M-5 (Yueshen28)	13.07 ± 0.19 bc	15.99 ± 0.73 e	11.91 ± 0.20 j	1.54 ± 0.03 g	5.24 ± 0.02 j
M-6 (Yueshen74)	12.90 ± 0.37 bc	13.03 ± 0.21 f	24.14 ± 0.56 d	2.86 ± 0.08 a	7.52 ± 0.04 g
M-7 (Da10)	11.50 ± 0.25 d	13.16 ± 0.52 f	23.66 ± 0.45 d	2.58 ± 0.07 c	9.44 ± 0.05 c
M-8 (Jinqaing63)	14.13 ± 0.46 b	48.49 ± 1.43 a	22.76 ± 0.13 e	2.23 ± 0.03 d	8.46 ± 0.05 de
M-9 (Yunguo1)	12.27 ± 0.53 cd	9.59 ± 0.91 g	17.13 ± 0.55 i	1.59 ± 0.01 g	5.82 ± 0.11 h
M-10 (Heizhenzhu)	6.20 ± 0.45 f	34.79 ± 0.22 b	27.29 ± 0.16 a	2.75 ± 0.04 b	11.15 ± 0.06 b
M-11 (Xuanguo1)	11.17 ± 0.54 d	47.39 ± 0.58 a	17.42 ± 0.41 h	1.87 ± 0.04 f	5.75 ± 0.15 hi
M-12 (Guiyou12)	15.83 ± 0.52 a	21.97 ± 0.07 d	18.17 ± 0.33 g	1.83 ± 0.06 f	5.59 ± 0.13 i
M-13 (Shansang)	14.12 ± 0.46 d	8.44 ± 0.15 gh	20.79 ± 0.07 f	2.18 ± 0.03 d	7.82 ± 0.05 f

The results are expressed as mean ± standard deviation (*n* = 3). Values in the same column with different letters are significant differences (*p* < 0.05). CA, citric acid; FW, fresh weight; GAE, gallic acid equivalents; DW, dry weight; RE, rutin equivalents; CE, cyanidin-3-*O*-glucoside equivalents.

**Table 2 foods-11-01252-t002:** Identification of phenolic compounds in the different mulberry samples.

Peak No.	Compound Name	RT (min)	Formula	[M + H]^+^/[M − H]^−^ (*m*/*z*)	MS^2^ (*m*/*z*)
1	Gallic acid	4.55	C_7_H_6_O_5_	[−]/169.0136	125, 124, 97
2	Protocatechuic acid	7.96	C_7_H_6_O_4_	[−]/153.0197	109, 108, 91
3	Cyanidin-3,5-diglucoside	9.60	C_27_H_31_O_16_	[+]/611.1301	449, 287
4	Protocatechuicaldehydea	9.61	C_7_H_6_O_3_	[−]/137.0237	107
5	p-Hydroxybenzoic acid	9.89	C_7_H_6_O_3_	[−]/137.0254	93
6	Aesculin	9.98	C_15_H_16_O_9_	[−]/339.0794	177, 133
7	Procyanidin B1	10.07	C_30_H_26_O_12_	[−]/577.1345	425, 407, 289
8	Procyanidin B3	10.44	C_30_H_26_O_12_	[−]/577.1364	425, 407, 289
9	Catechin	10.78	C_15_H_14_O_6_	[−]/289.0790	245, 205, 179, 153
10	Chlorogenic acid	10.98	C_16_H_18_O_9_	[−]/353.0866	191, 179, 173
11	Cryptochlorogenic acid	11.31	C_16_H_18_O_9_	[−]/353.0877	191, 179, 135
12	Aesculetin ^a^	11.39	C_9_H_6_O_4_	[−]/177.0266	133
13	Vanillic acid	11.42	C_8_H_8_O_4_	[−]/167.0360	167, 108
14	Caffeic acid	11.53	C_9_H_8_O_4_	[−]/179.0343	135, 107
15	Fraxin ^a^	11.92	C_16_H_18_O_10_	[−]/369.0820	206, 160, 112
16	Syringic acid	12.29	C_9_H_10_O_5_	[−]/197.0528	182, 167, 153, 138, 123
17	Pelargonidin-3-*O*-glucoside	12.46	C_21_H_21_O_10_	[+]/432.9823	271
18	Epicatechin	12.63	C_15_H_14_O_6_	[−]/289.0790	245
19	Dihydromyricetin ^a^	12.73	C_15_H_12_O_8_	[−]/319.0532	193
20	1,5-Dicaffeoylquinic acid	13.12	C_25_H_24_O_12_	[−]/515.1428	353, 191, 179
21	Vanillin^a^	13.46	C_8_H_8_O_3_	[−]/151.0473	136, 123, 108
22	Perillyl alcohol ^a^	13.49	C_10_H_16_O	[−]/151.1122	135
23	P-Hydroxycinnamic acid	13.90	C_7_H_6_O_3_	[−]/137.0254	93
24	Umbelliferone ^a^	14.53	C_9_H_6_O_3_	[+]/162.0316	107
25	Ferulic acid	15.26	C_10_H_10_O_4_	[−]/193.0500	149, 178, 134
26	trans-Piceid	15.31	C_20_H_22_O_8_	[−]/389.1235	227
27	Sinapic acid ^a^	15.45	C_11_H_12_O	[−]/223.0605	179
28	Ellagic acid	15.53	C_14_H_6_O_8_	[−]/300.9983	283, 257, 229, 185
29	Rutin	15.54	C_27_H_30_O_16_	[−]/609.1485	301, 300
30	Delphinidin-3-*O*-glucoside	15.74	C_21_H_21_O_12_	[+]/465.1033	303
31	Taxifolin	15.81	C_15_H_12_O_7_	[−]/303.0583	285, 177, 125
32	Quercetin-3-*O*-galactoside	15.87	C_21_H_20_O_12_	[−]/463.0910	301, 300
33	Pelargonidin	16.09	C_15_H_11_O_5_	[+]/271.5795	197, 121
34	Piceatannol	16.48	C_14_H_12_O_4_	[−]/243.0735	159
35	Cyanidin-3-*O*-rutinoside	16.88	C_27_H_31_O_15_	[+]/595.1351	287
36	Kaempferol-3-*O*-rutinoside	16.89	C_27_H_30_O_15_	[−]/593.1532	327, 285, 284
37	Coniferaldehyde ^a^	17.00	C_10_H_10_O_3_	[+]/179.0708	147
38	Salicylic acid ^a^	17.06	C_7_H_6_O_3_	[−]/137.0237	93
39	Narcissin ^a^	17.22	C_28_H_32_O_16_	[−]/623.1690	315
40	Astragalin	17.33	C_21_H_20_O_11_	[−]/447.0926	285, 284
41	Prunin ^a^	17.55	C_21_H_22_O_10_	[−]/433.1133	271
42	Isorhamnetin-3-*O*-glucoside	17.69	C_22_H_22_O_12_	[−]/477.1032	315, 300
43	Hesperidin ^a^	17.95	C_28_H_34_O_15_	[−]/609.1818	301, 125
44	Aromadendrin ^a^	17.99	C_15_H_12_O_6_	[−]/287.0633	259, 243
45	Phlorizin	18.58	C_21_H_24_O_10_	[−]/435.1290	273
46	Resveratrol	19.10	C_14_H_12_O_3_	[−]/227.0786	185
47	Trilobatin ^a^	19.79	C_21_H_24_O_10_	[−]/435.1290	273
48	Eriodictyol	20.64	C_15_H_12_O_6_	[−]/287.0633	287, 151
49	trans-Cinnamic acid	21.15	C_9_H_8_O_2_	[−]/147.0524	119, 103
50	Quercetin	21.22	C_15_H_10_O_7_	[−]/301.0360	273, 257, 178, 151
51	Luteolin	21.27	C_15_H_10_O_6_	[−]/285.0477	241, 175, 217, 199
52	Morin	21.28	C_15_H_10_O_7_	[−]/301.0426	271, 257, 193, 178
53	Butein ^a^	23.10	C_15_H_12_O_5_	[−]/271.0684	135
54	Phloretin	23.56	C_15_H_14_O_5_	[−]/273.0841	167
55	Kaempferol	24.12	C_15_H_10_O_6_	[−]/284.0322	285, 257, 185, 169, 151

^a^ Compounds identified in the mulberry samples for the first time.

**Table 3 foods-11-01252-t003:** Phenolic compound contents (mg/kg DW) in the mulberry samples of different cultivars.

Compound Name	Mulberry Sample												
	M-1	M-2	M-3	M-4	M-5	M-6	M-7	M-8	M-9	M-10	M-11	M-12	M-13
Phenolic acids													
Gallic acid	1.66	1.32	0.17	1.68	0.88	0.74	0.64	0.06	0.81	1.84	0.61	0.66	0.71
Protocatechuic acid	34.60	43.30	14.47	101.74	22.60	42.01	36.71	14.32	36.59	46.99	25.57	18.45	19.55
p-Hydroxybenzoic acid	0.97	1.44	0.71	4.71	0.80	0.93	1.18	0.53	1.52	2.24	0.78	0.89	0.56
Chlorogenic acid	138.35	192.16	56.94	306.01	110.01	206.35	177.28	74.71	200.16	160.73	123.69	78.75	84.47
Cryptochlorogenic acid	193.24	259.97	83.55	422.18	118.84	109.61	229.75	122.89	81.50	212.63	150.13	91.77	98.32
Vanillic acid	0.24	0.75	0.16	1.02	0.20	0.38	0.44	0.14	0.40	0.60	0.89	0.19	0.24
Caffeic acid	1.03	3.73	0.80	6.52	1.07	2.26	1.77	0.94	3.04	5.27	0.67	0.66	0.95
Syringic acid	0.16	0.13	0.04	0.35	0.04	0.10	0.08	0.01	0.05	0.31	0.06	0.03	0.02
4-Hydroxycinnamic acid	1.32	3.77	2.33	3.28	0.98	3.59	1.29	0.90	0.52	3.70	0.40	0.86	1.77
Ferulic acid	0.20	0.51	0.32	0.52	0.20	0.51	0.37	0.09	0.24	0.56	0.24	0.13	0.35
Sinapic acid	0.34	0.72	0.27	0.51	0.27	1.13	1.03	0.13	0.44	1.05	0.78	0.13	0.24
Ellagic acid	5.15	1.40	1.15	2.95	1.72	2.73	2.77	1.09	4.19	6.40	2.69	1.27	1.90
Salicylic acid	0.20	0.36	0.16	0.84	0.25	0.29	0.27	0.07	1.17	0.37	0.58	0.19	0.30
trans-Cinnamic acid	0.12	0.14	0.19	0.14	0.25	0.15	0.22	0.05	0.11	0.27	0.08	0.15	1.28
Total phenolic acids													
	377.58	509.7	161.26	852.45	258.11	370.78	453.8	215.93	330.74	442.96	307.17	194.13	210.66
Flavonols													
Dihydromyricetin	5.10	7.53	2.09	14.24	4.68	6.90	6.46	2.16	5.36	5.87	3.93	4.34	3.44
Rutin	42.17	51.93	58.74	139.31	39.22	54.60	41.56	100.88	38.96	53.51	27.88	18.13	69.07
Taxifolin	9.84	18.70	3.11	34.06	14.70	13.28	5.86	8.05	4.67	30.74	5.27	4.76	9.49
Quercetin-3-O-galactoside	15.00	20.24	3.36	52.13	4.33	21.76	11.12	9.65	8.09	14.56	12.54	2.73	4.02
Kaempferol-3-O-rutinoside	47.07	52.54	10.02	107.64	22.80	47.84	58.24	21.33	51.20	55.24	31.14	18.60	10.39
Narcissin	1.37	1.02	0.16	2.18	0.35	0.62	0.80	0.07	0.93	1.58	0.87	0.24	0.43
Astragalin	11.72	26.10	4.95	29.59	4.53	22.89	7.76	2.58	16.46	12.05	6.11	3.89	4.40
Isorhamnetin-3-O-glucoside	0.04	0.07	0.03	0.16	0.02	0.07	0.05	0.02	0.03	0.04	0.03	0.02	0.03
Aromadendrin	2.75	2.35	0.56	5.06	2.41	1.59	1.05	2.18	1.42	3.68	0.84	1.53	1.01
Quercetin	31.06	36.09	6.89	56.52	19.93	33.98	35.14	5.41	31.93	32.18	21.35	16.87	16.75
Kaempferol	1.66	0.79	0.15	1.38	1.50	0.79	1.47	0.09	1.33	0.92	0.54	0.78	0.53
Flavanones													
Prunin	4.86	9.35	1.25	11.49	9.72	10.91	9.57	5.71	5.63	12.08	7.64	7.01	3.29
Hesperidin	1.37	2.07	0.29	3.84	0.59	0.83	1.39	0.40	1.26	1.44	0.43	0.44	0.20
Eriodictyol	10.79	3.30	0.29	16.65	3.88	2.88	3.28	0.96	1.23	36.68	0.78	2.18	3.73
Butein	1.55	0.20	0.03	0.52	0.45	0.25	0.66	0.06	0.84	2.21	0.24	0.35	0.14
Flavanols													
Catechin	0.26	0.19	0.11	0.37	0.11	0.15	0.09	0.11	0.19	0.45	0.22	0.11	0.25
Epicatechin	0.21	0.54	0.08	0.41	0.07	0.45	0.07	0.11	0.40	0.28	0.19	0.05	0.04
Flavones													
Luteolin	20.54	10.34	1.66	22.59	11.87	11.31	9.21	0.98	6.25	51.83	4.94	10.60	7.81
Morin	24.17	28.07	6.03	45.67	14.37	26.84	26.57	5.14	24.15	25.21	15.85	12.04	12.44
Dihydrochalcones													
Phlorizin	6.87	10.30	2.16	13.33	5.20	9.54	6.46	1.65	3.04	12.34	1.40	4.46	2.90
Trilobatin	7.49	11.96	2.43	12.08	2.00	14.16	5.28	0.34	2.67	13.70	1.81	1.54	1.08
Phloretin	1.02	0.46	0.04	0.49	0.28	0.32	0.24	0.04	0.27	3.22	0.14	0.17	0.17
Total flavonoids													
	246.94	294.16	104.45	569.70	163.01	281.95	232.33	167.92	206.32	369.83	144.14	110.85	151.63
Anthocyanins													
Cyanidin-3,5-diglucoside	12.83	9.61	5.87	72.75	6.04	7.35	11.59	6.50	5.46	23.12	5.07	5.76	6.39
Pelargonidin-3-O-glucoside	–	–	0.12	–	0.46	–	0.68	–	–	–	–	–	–
Delphinidin-3-O-glucoside	27.65	11.78	6.89	74.22	52.17	9.94	5.93	11.93	3.47	14.83	9.77	6.41	2.36
Pelargonidin	0.31	0.24	0.10	0.43	0.28	0.33	0.42	–	0.32	0.31	0.16	0.08	0.08
Cyanidin-3-O-rutinoside	47.47	54.41	10.20	92.06	34.98	44.94	66.17	10.01	56.04	60.59	30.93	21.80	12.34
Total anthocyanins													
	88.26	76.51	23.17	239.46	93.93	62.56	84.79	28.43	65.30	98.85	45.94	34.06	21.17
Other phenolic compounds													
Procyanidin B1	0.05	0.09	0.18	0.34	0.02	0.03	0.02	0.09	0.02	0.05	0.05	0.02	0.05
Procyanidin B3	0.04	0.01	0.10	0.04	0.01	0.01	0.01	0.04	0.03	0.05	0.01	0.01	0.02
Fraxin	0.17	0.45	0.06	0.36	0.07	0.48	0.13	0.02	0.51	0.29	0.41	0.06	0.06
trans-Piceid	1.46	0.73	0.26	2.65	0.65	0.84	1.16	1.16	2.02	3.01	1.05	0.46	0.50
Resveratrol	2.51	1.23	0.18	3.74	1.09	1.10	2.08	0.25	1.72	5.18	0.76	0.56	0.85
Total individual phenolic compounds													
	717.85	883.89	292.16	1672.21	517.15	718.11	774.61	404.17	607.09	921.05	500.08	340.53	385.58

**Table 4 foods-11-01252-t004:** Correlations between total phenolic contents, total flavonoid contents, total anthocyanin contents and antioxidant activities.

	TPC	TFC	TAC	DPPH	ABTS	FRAP
TPC	1					
TFC	0.754 **	1				
TAC	0.824 **	0.952 **	1			
DPPH	0.882 **	0.868 **	0.884 **	1		
ABTS	0.960 **	0.935 **	0.626 *	0.948 **	1	
FRAP	0.632 *	0.626 *	0.830 **	0.823 **	0.775 **	1

*, *p* < 0.05; **, *p* < 0.01.

## Data Availability

Data is contained within the article.

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
