# Peer review of "Evaluation of Different Black Mulberry Fruits (Morus nigra L.) Based on Phenolic Compounds and Antioxidant Activity"

_foods, 2022, doi:10.3390/foods11091252_

Round 1

Reviewer 1 Report

Dear Editor,

Thank you for the invitation. I didn't find any novelty in the study, though it can be improved. Kindy do find  the suggestion below

  1. The title needs to be a little shortened or needs to be more specific.
  2. Authors need to mention by ABTS and DPPH both methods were used in the study. Although both work on a different principle. Is there any specific reason for the same?
  3. Why were these all Mulberries selected for the study? Is there any specific reason? If you look at previous studies all work is done previously. Therefore the novelty of the work is questionable.
  4. Could the authors indicate the sample was taken as fresh or frozen samples? It's not clear. Storage conditions also need to be considered.
  5. PCA plot and Hirarical plot need to be properly explained and need to be compared with previous studies reported if any.

I still keep my decision reserved. Therefore, I will go with the editor's decision.

Reviewer 2 Report

Dear Editor, dear Authors,

After reviewing the paper entitled "Evaluation of different black mulberry fruits (Morus nigra L.) based on phenolic compounds and antioxidant activity", I can suggest a major revision.

These are my comments for the Authors:

  • 2.1. Section: please indicate maturity, a season of picking, etc. for all fruits. Did fruits wash before storage - indicate preparation process and storage conditions here.
  • 2.4. section indicate freezing process parameters
  • The conclusion part is very superficial. Please, add all remarks and a summary of the obtained results in this part with all advantages and disadvantages, further perspectives, etc. 
  • The phytochemical characterization is very superficial. The Folin–Ciocalteau method has been shown not to be a good method to express Total phenolic content because it overestimates these compounds, by also reacting with other non-polyphenolic antioxidant compounds. The same goes for other assays used in the manuscript, which have become obsolete (DPPH, ABTS). Please, expand the discussion with an explanation of this hypothesis and how the Authors think they can overcome these shortcomings of the used methods. 

Reviewer 3 Report

Article

Evaluation of different black mulberry fruits (Morus nigra L.) 2 based on phenolic compounds and antioxidant activity

Journal: Foods

Comments to Author,

Scientific names should be italic.

First it should be Morus alba and then  M. alba.  Please correct it thoroughly.

Line number 15: complete this statement, something is missing. Or recheck, clearly that what you want to say.

Line number 37: can you please differentiate these three varieties on the basis of their content in percentage?

Line number 40: How much yield cultivated? And which variety? Which is abundantly available? And reason?

Line number 47: Need to write at the end of introduction, what is the purpose of your study.

Line number 77: sonicated? At which capacity?

Line number 225: how similar to previous report. Can you please elaborate previous report with present study results?

Line number 241: stable products, for example?

Line number 254: Need to write percentage of lower and higher value of mulberry, either talk about white or black

Line number 262: what was previous finding? Compare your results. And show to reader that what difference was.

Line number 302: mulberry sample, white mulberry sample or black? Or Overall? Don’t make confusion please

Line number 323: Overall research is based on evaluation, after that what will be the future recommendation for new researchers regarding t your research. Need to add at the end of conclusion
